# Identification of *Vitis* Cultivars, Rootstocks, and Species Expressing Resistance to a *Planococcus* Mealybug

**DOI:** 10.3390/insects11020086

**Published:** 2020-01-28

**Authors:** Rachel P. Naegele, Peter Cousins, Kent M. Daane

**Affiliations:** 1USDA ARS, San Joaquin Valley Agricultural Sciences Center, Parlier, CA 93611, USA; 2E. & J. Gallo Winery, Modesto, CA 95354, USA; Peter.Cousins@ejgallo.com; 3Department of Environmental Science, Policy, and Management, University of California Berkeley, Berkeley, CA 94720-3114, USA; kdaane@ucanr.edu

**Keywords:** host plant resistance, pest management, *Planococcus ficus*, vineyard

## Abstract

Mealybugs cause economic loss to vineyards through physical damage, fouling fruit and leaves with honeydew, and the transmission of viruses. *Planococcus ficus* is one of several mealybug species in vineyards, and one that causes economic damage over a relatively large global range. To develop novel management tools, host resistance to *P. ficus*, which has not previously been identified for any grape cultivars, was studied. Ten grape lines (species, cultivars, and rootstocks) were evaluated for *P. ficus* resistance across two separate potted plant assays. Significant differences were detected among cultivars and rootstocks in the recorded number of *P. ficus* juveniles, adults, and egg sacs. Cabernet Sauvignon and Chardonnay were two of the most favorable grape cultivars for mealybug population growth, whereas rootstocks IAC 572, 10-17A, and RS-3 all demonstrated some level of resistance. Southern fire ant (*Solenopsis xyloni*) was positively associated with mealybug populations, but did not have a negative effect on the observed presence of other arthropod species including potential predators.

## 1. Introduction

Grapes have a long history of cultivation and breeding for a wide range of soils, climates and commodities (e.g., table grapes for fresh consumption and processed grapes that are dried into raisins or pressed for grape juice or wine) [1]. Although there have been numerous studies on the development of resistant cultivars to fungal and viral pathogens [2,3,4] and nematodes [5], there has been little work on grape cultivars that are resistant to key arthropod pests, with the exception of grape phylloxera [6]. Globally, mealybugs (Hemiptera: Coccoidea: Pseudococcidae) are one of the more important arthropod pests in vineyards [1]**,** and economic losses resulting from mealybugs have dramatically increased in the past decades, in part as a result of globalization [7]**,** despite the fact that many countries impose regulations on the movement of vine material [8].

Vineyard mealybugs are phloem-feeding pests that can cause economic loss through feeding damage to leaves, resulting in reduced photosynthetic capability, and the excretion of carbohydrate-rich honeydew that can further foul the leaves, stems, and fruit and lead to the accumulation of sooty molds [9,10] (Figure 1). In addition to losses attributed directly to feeding, mealybugs can transmit grapevine leafroll associated viruses (GLRaVs), resulting in grape leafroll disease (GLD) [11,12] (Figure 1), which has been estimated to cost growers between $12,106 to $91,623 USD per acre annually in California [13]. Of that expenditure, mealybug control costs were estimated to range from $50 USD per acre for vineyards with low mealybug population densities, and up to $500 USD per acre for vineyards with moderate to large population densities [13]. At least 10 mealybug species have been identified globally that have risen to the level of economic pest in vineyards [9]. *Planococcus ficus* (Signoret) is one of the most important vineyard mealybugs that has a global distribution [14], is a known vector of GLRaVs [15,16,17,18], and has become the primary pest in California vineyards [19]. 

Integrated pest management (IPM) systems are integral for mealybug management primarily in the table and wine grape markets, and include cultural practices, such as cluster thinning and bark stripping; however, most farmers still rely on chemical controls to minimize exposure of the clusters to mealybugs [19,20]. More sustainable tools for *P. ficus* control include mating disruption, which is currently being used or tested worldwide as an alternative or complement to insecticide sprays [21,22,23,24]. Biological controls are another tool to help suppress *P. ficus* populations, with a number of predators that attack mealybugs, including the mealybug destroyer, *Cryptolaemus montrouzieri* Mulsant, lacewings (e.g., *Chrysoperla* spp.), and cecidomyiid flies (predaceous midges such as *Diadiplosis koebelei* (Koebele)) [25,26,27,28,29]. Most successful biological control programs rely primarily on encrytid parasitoids [30], such as *Anagyrus pseudococci* (Girault), a parasitoid of *P. ficus* and other related mealybugs [26,29,31,32,33]. Even in organic vineyards, natural enemies may not provide complete control—ants have been shown to disrupt mealybug biological control in vineyards [33,34,35,36] and *P. ficus* can find refuge from some natural enemy species under the vines bark [37]. Some mealybug species can also overwinter on the roots of grapevines, and rootstocks with resistance to mealybug would be a valuable tool as part of IPM. For these reasons, additional control tools should still be investigated. 

Host resistance to *P. ficus* has not yet been developed or even investigated for grape. Classic development of plant host resistance to insects typically occurs through antixenosis or antibiosis [38]. In antibiosis, the host adversely effects the insect resulting in increased mortality and reduced fecundity or longevity, whereas antixenosis affects the behavior of the insect, resulting in migration to a more favorable host [39]. Resistance to other pests such as phylloxera (*Daktulosphaira* spp.) and nematodes (*Meloidogyne* spp.) have been identified in grape, primarily in native American species, which may serve as a useful source of resistance to other pests [5,40,41]. Few sources of plant host resistance to mealybugs have been identified, although antibiotic components of resistance were reported in cassava cultivars to the cassava mealybug, *Phenacoccus manihoti* Matile-Ferrero, reducing the insect’s reproductive capacity [42]. Similarly, antiobiosis resistance was described for a grape rootstock to the citrus mealybug, *Planococcus citri* (Risso), that had a reduction in the number of viable offspring compared to susceptible cultivars [43]**,** and these results were later reproduced with the pineapple mealybug, *Dysmicoccus brevipes* (Cockerell) [44]. Our aim was to evaluate the potential of grape rootstocks to impart resistance to *P. ficus* to improve vineyard IPM. 

## 2. Materials and Methods

### 2.1. Germplasm Evaluation

Own-rooted cuttings were collected from mature field-grown grapevines, including rootstocks and species at the San Joaquin Valley Agricultural Sciences Center (SJVASC), Parlier, CA, or the University of California’s Kearney Agricultural Research and Extension Center (KARE), Parlier, CA (Table 1). Plant material was selected on the basis of known resistance to nematodes and suspected resistance to mealybugs, as well as other agronomic traits of value. Ten replicate rooted plants of a similar size and age were transplanted into round 15.24 × 30.48 cm^2^ black tree-pots (CP612R, Stuewe and Sons Inc, Tangent, OR), placed into a completely randomized design, and maintained in a screened cage (approximately 2.5 × 2.5 × 1.8 m) at SJVASC. Potted plants were treated every 2 weeks with sulfur to control powdery mildew, but did not receive any other insecticide treatments. Plants were watered as needed.

*Planococcus ficus* used was from an established colony reared on butternut squash at a KARE insectary; the material originated from *P. ficus*-infested vines in Fresno County. To inoculate potted vines, a single egg sac (approximately 50–500 eggs) was placed onto a 60 mm piece of filter paper that was then attached to each grape plant by wrapping the filter paper around the base of the stem and securing it with a stapler. One week later, a second egg sac was placed onto each plant using the same method. Plants were inoculated in June of 2017.

Plants were evaluated every 1–2 weeks for a total of 12 weeks, for the number of mealybugs (recorded as juveniles, or adult females) and ovisacs counted during a 1 minute rating period. A pre-existing southern fire ant colony was located near (<2 m) the study, and the total number of ants were recorded for each plant. Possible *P. ficus* predators, including lacewings, spiders, robber flies, and other species, were counted as presence/absence on the basis of observation of the animal or parasitized mealybugs. After the trial was established, southern fire ants, *Solenopsis xyloni* McCook, were observed tending mealybugs and were not considered predatory. The entire experiment was repeated using a separate cage, approximately 1 week after the initiation of the first experiment to repeat the initial experiment and confirm results. 

### 2.2. Cultivar Evaluation

A second experiment was conducted to determine differences in the number of mealybugs among cultivars, with data recorded including different mealybug life stages. Own-rooted cuttings were generated from mature field-grown non symptomatic grapevines at the SJVASC (Table 2). Ten replicate rooted plants were transplanted into black tree pots (CP612R) and placed into similar sized pots buried into the ground with an 8 cm block in the bottom to raise the internal pot height (7.62 × 7.62 cm^2^) (Figure 2). Two of the cultivars were only represented by five plants due to their availability (Table 2). To minimize the presence and impact of predators and parasitoids, each vine was covered with a paint strainer bag (mesh size of approximately 200 microns). Plants were treated every 2 weeks with sulfur to control powdery mildew, but did not receive any insecticide treatments during the trial or 8 months prior to the start of the experiment. Plants were watered as needed. Although rare in comparison to mealybugs, visible predators, caterpillars, and other non-target potential predatory insects (not including ants) were removed from plants when observed.

For inoculations, 200 crawlers (first or second instar) were transferred to 60 mm filter paper using a paintbrush, placed onto the base of each plant. A second set of 200 crawlers was placed onto each plant using the same method 1 week later, for a total of 400 crawlers per plant. The total number of mealybugs were counted on each plant every 2 weeks during a 1 min timed search [46] recording mealybug numbers and their developmental stage (first, second, and third instars; adults; and ovisacs). A pre-existing southern fire ant population was located near the study, and the total number of ants were recorded for each plant on each sample date. Plant health was also monitored using a 1–5 scale with 1 (dead plant or <90% defoliation or necrosis), 2 (chlorosis, necrosis, or defoliation on 70% to 90% of the plant), 3 (chlorosis, necrosis, or defoliation on 35% to 70%), 4 (chlorosis, necrosis, or defoliation on 10% to 35% of the plant), and 5 being completely healthy (chlorosis or necrosis on <10% of the plant), to describe how the mealybug populations were affecting the overall appearance of each cultivar. Plants were inoculated with mealybugs in early July in 2018.

### 2.3. Data Analysis

For each plant, an area under the insect growth curve (AIGC) value similar to the calculated aphid days (CAD) [47] was determined on the basis of the area under the disease progress curve (AUDPC) calculation by Shaner and Finney [48]. In brief, the number reflects insect population growth on each plant by accounting for the rate of change between sample dates on the basis of:$$AIGC=\sum_{i=1}^{N_{i}-l}\frac{y_{i}+y_{i+1}}{2}*(t_{i+1}-t_{i})$$
where, for each rating period, the number of insects observed (y_i_) and the difference from the next rating period (y_i+1_) are averaged and multiplied by the amount of time (t) between the rating periods. The sum of these calculations for the total number of observations (N) is the AIGC. 

Results are presented as sample means (±SEM). For each plant, an area under the insect growth curve (AIGC) value was determined on the basis of the AUDPC calculation by Shaner and Finney for both ants and vine mealybugs [48]. The number of third instars to adults were combined for analyses (e.g., first instars and third instars to adults were analyzed separately). Data were compared for each line and cage using LSMeans implemented within SAS statistical analysis software v. 9.3 (Cary, NC) using the function PROC MIXED. For the cage study, data were square root transformed (mealybugs and ants) or log 10 transformed (predators) prior to analyses to improve normality, and means were separated using Tukey’s honest significant difference at *p* < 0.05. Pearson’s correlation coefficient was calculating using PROC Corr implemented within SAS. For the cultivar study, data were log transformed (vine mealybugs and ants) prior to analyses to improve normality, means were separated using Tukey’s Honest Significant Difference at *p* < 0.05. Pearson’s Correlation Coefficient was calculating using PROC Corr implemented within SAS. Plant health was evaluated using non-parametric comparisons for all pairs using the Dunn method for joint ranking implemented within JMP statistical software v. 12.0.1 (Cary, NC, USA).

## 3. Results

### Annual Generations and Seasonal Development

For the first cage experiment, significant differences were detected among cultivars (*p* < 0.0001), cages (*p =* 0.0019), and line by cage interactions (*p =* 0.0084) for mealybug population growth. Cabernet Sauvignon, the susceptible control, consistently had higher numbers of mealybugs and ants throughout the experiment compared with any of the other materials evaluated (Table 3). Population growth (AIGC values) was lower, on average, in the second run of the experiment compared to the first experiment. No significant differences were detected among the rootstock cultivars or wild species in the first run, and only minor differences were found in the second (*p* = 0.05). It should be noted that counts for juvenile and adult female mealybugs were combined in this study. Juveniles (first and second instars) are more difficult to detect, as they are quite small and will hide in crevices, making individual rating dates quite variable. The number of ants detected per rootstock and wild species had a significant cultivar (*p* < 0.0001) and cage (*p* < 0.0001) effect, but no significant interaction was detected between cultivar and cage. The average frequency of predators detected was significantly different among cultivars (*p* < 0.0001), but not by cage or the interaction between cultivar and cage. A greater numbers of ants was associated with higher numbers of mealybugs across all lines evaluated (*r* = 0.62115, *p* < 0.0001); however, ant density was not associated with predator presence (*r* = 0.17420, *p* = 0.0581). Predator frequency had a small but significant positive association (*r* = 0.24355, *p* = 0.0076) with mealybugs. Therefore, the presence of ants and predators, which was not expected when the project was initiated, was likely related to the presence of mealybugs, but must be accounted for in future trials.

For the cultivar evaluation experiment, significant differences were detected among cultivars for first instars (*p* < 0.0001) and third instars and adults (*p* = 0.0007). Chardonnay had the greatest number of mealybugs (juveniles, adults, and ovisacs) and was significantly different compared to both IAC 572 and RS-3 rootstocks (Table 4). Most of the commercially available scion cultivars were not significantly different from each other. Valley Pearl had a lower number of crawlers compared to the other cultivars, but was not significantly different in the number of adult female mealybugs visible. Rootstocks IAC 572 and RS-3 both had fewer mealybugs (juveniles and adults) than cultivars Chardonnay, Autumn King, and Cabernet Sauvignon, but high variability in mealybug populations were observed within scion cultivars. Strong correlations were observed between the number of ants detected and mealybug populations (*r* = 0.37087, *p* = 0.0001 and *r* = 0.4864, *p* < 0.0001 for crawlers and adults, respectively). Plant health was similar among the cultivars, with the only significant differences being with IAC 572 against Cabernet Sauvignon and Valley Pearl, and Cabernet Sauvignon compared to Autumn King.

## 4. Discussion

*Planococcus ficus* is a serious insect pest of grapes with no management tools that provide complete control [14,20,23,49,50,51,52]. We evaluated 10 grape cultivars, rootstocks, and species for their relative resistance to *P. ficus* population growth in field-based cage studies. Each of the rootstocks evaluated showed reduced mealybug population numbers compared to *V. vinifera* controls (cv. Cabernet Sauvignon and Chardonnay), and we suggest the reduced population growth could result from some level of antibiosis or antixenosis resistance mechanisms. Female mealybugs, while sessile when adults, can travel several feet or more to find a host during early stages of development. Because crawlers were able to move among plants in the first experiment, we cannot omit the possibility that antixenosis was contributing to our results. However, this would also be the case in a vineyard setting, where mealybugs could crawl between vines. In contrast to previous studies by Bertin et al. [44] and Filho et al. [43], IAC 572 did have some mealybugs visible throughout both of the studies, suggesting that viable offspring were produced, though in low numbers. This could be, in part, due to differences in the three mealybug species in host preference and reproduction methods. RS-3, which had been suspected to protect roots against vine mealybug [53], showed few crawlers and adults throughout the study, suggesting that the resistance mechanism employed may be antibiosis. However, ants and predators were unexpectedly found in these trials, and although their presence was likely directly related to mealybug densities, they may also have host preferences which could influence mealybug populations and may contribute to resistance responses. These data suggest that sources of resistance to vine mealybug do exist, and that there may be differences in mechanisms and levels of resistance among *Vitis* spp. These resistance mechanisms could be limited to mealybug suitability, but may also affect mealybug relationships with their ant caretakers by changing nutrient composition of honeydew. Ants affect natural enemy effectiveness, although their impact depends on the ant and natural enemy species [34,35,54,55]. Surprisingly, in our study ant populations were associated with higher mealybug numbers, but had little effect on presence/absence of natural enemies.

Though all *V. vinifera* cultivars appear to be susceptible to vine mealybug, scion variability exists. In our results, table grape cultivars Valley Pearl and Flame Seedless had numerically fewer adult mealybugs and egg sacs over time compared to the wine grape cultivars Chardonnay and Cabernet Sauvignon and the table grape cultivar Autumn King. Though this was only statistically significant when comparing cultivars Chardonnay and Autumn King with cultivar Valley Pearl. This is similar to results from previous studies evaluating mealybug resistance in cassava, mango, and buffalo grass where cultivar differences were observed [42,56]. Using resistant rootstocks with scion cultivars that have reduced susceptibility to mealybug could help reduce mealybug populations living under the bark or overwintering on roots that systemic insecticides have trouble targeting.

In summary, we identified at least two potential sources of resistance to vine mealybug under potted plant conditions in a semi-natural environment. The commercially available though not widely used rootstocks IAC 572 and RS-3 may be useful in grape growing regions with high mealybug pressure as part of an IPM program. Further studies to confirm and identify additional sources of resistance and determine mechanisms contributing to this potential resistance without the presence of ants or mealybug predators are needed. 

## 5. Conclusions

*Planococcus ficus* is one of several mealybug species found in grape vineyards globally. Resistant grape cultivars, which are an important component of IPM, are not available to manage this insect pests. Evaluation of grape rootstocks and cultivars identified differences in mealybug population growth. Both juvenile and adult female mealybug, and southern fire ant populations were lower on rootstocks than on cultivated varieties. Because of the variability in mealybug growth even on the rootstocks, it is likely that there are cultivar-specific mechanisms contributing to mealybug resistance. These mechanisms could be physical or chemical features that affect feeding and host attractiveness to mealybugs. Although several ant species are associated with *P. ficus*, the specific effect of each species on mealybug population growth has not been evaluated. The presence of ants was correlated with higher numbers of mealybugs, but not the absence of mealybug predators. Here, we demonstrated the existence of potential mealybug resistance in *Vitis* spp., and identified rootstocks useful for breeding and IPM. Follow-up studies should include multi-year evaluations of these rootstocks in a vineyard setting under high and low vine mealybug pressure and determine their effectiveness without ants or mealybug predators. 

## Figures and Tables

**Figure 1 insects-11-00086-f001:**
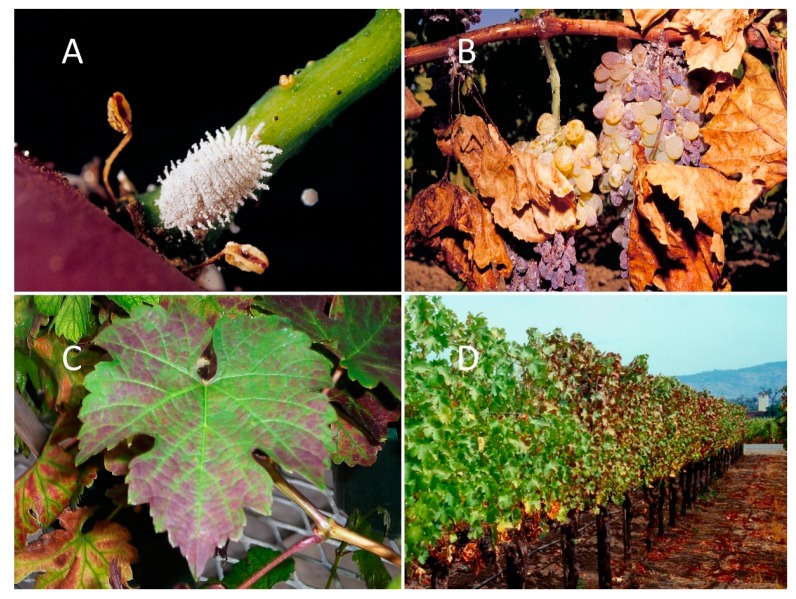
Globally, mealybugs have become some of the more important vineyard pests; shown here (**A**) an adult *Planococcus ficus* on a grape berry petiole; (**B**) direct damage from mealybugs feeding on grape leaves, showing defoliation, and fruit clusters, showing berry damage and raisining (drying); (**C**) a single leaf showing grape leafroll disease (GLD) on a red-cultivar wine grape caused by grape leafroll associated viruses transmitted by mealybugs; and (**D**) a GLD-infested vine row.

**Figure 2 insects-11-00086-f002:**
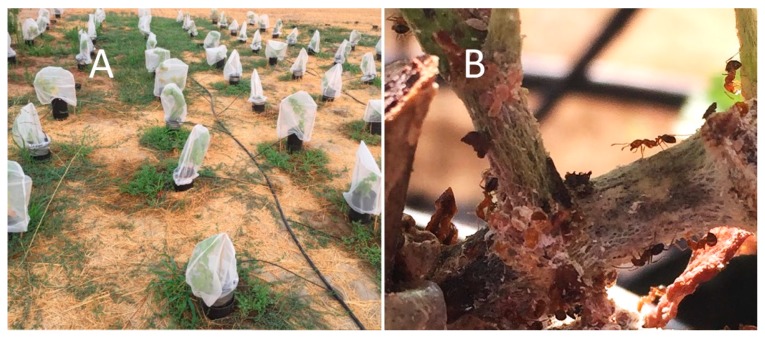
(**A**) Field design for the cultivar evaluation study testing seven *Vitis* lines for resistance to *Planococcus ficus* and (**B**) southern fire ants, *Solenopsis xyloni*, tending mealybugs in the trial, which become an inadvertent but potentially important aspect of mealybug response to *Vitis* cultivars.

**Table 1 insects-11-00086-t001:** Grape germplasm evaluated for resistance to vine mealybug in cage experiment.

Cultivar	*Vitis* Species	Features ^1^
USDA 1-2	*V. champinii*	Nematode resistance
PCO-349-11	Interspecific hybrid	Nematode resistance
IAC 572	*V. carabaea x* 101-14	Citrus mealybug resistance
10-17A	Interspecific hybrid	Nematode resistance
Australis ^2^	*V. longii*	Phylloxera resistance
Cabernet Sauvignon	*V. vinifera*	Wine grape control

^1^ Special characteristics (insect resistance) of each genotype selected. ^2^ Australis is a cultivar name [45].

**Table 2 insects-11-00086-t002:** Grape germplasm evaluated for resistance to vine mealybug in cultivar evaluation experiment.

Cultivar	No. Plants ^1^	Species	Features
Autumn King	10	*V. vinifera*	table grape control
Cabernet Sauvignon	10	*V. vinifera*	wine grape control
IAC 572	10	Interspecific hybrid	*Planococcus citri* and *Dysmicoccus brevipes* resistance
RS-3	5	Interspecific hybrid	mealybug resistance (anecdotal) ^2^
Flame Seedless	5	*V. vinifera*	table grape
Chardonnay	10	*V. vinifera*	wine grape
Valley Pearl	10	*V. vinifera*	table grape

^1^ Number of plants included in the study and used for analyses. ^2^ Based on observations by Dr. M. McKenry (personal communication).

**Table 3 insects-11-00086-t003:** Vine mealybug and ant population growth on grapevines evaluated in cage experiment.

Cultivar	Trial ^1^	Mealybug AIGC ^2^		Ant AIGC		Predators ^3^	
10-17A	1	192	B^4^	184.8	C	68%	A
Cabernet Sauvignon	1	733.1	A	869.4	A	48%	AB
IAC 572	1	89.5	B	242.9	BC	23%	C
PCO-349	1	151.75	B	379.4	B	30%	BC
Australis	1	205.75	B	291.2	BC	25%	C
USDA 1-2	1	99.8	B	340.9	BC	33%	BC
10=17A	2	127.75	B	19.15	B	45%	AB
Cabernet Sauvignon	2	1026.1	A	549.0	A	53%	A
IAC 572	2	8.8	C	27.5	B	33%	ABC
PCO-349	2	53	BC	36.0	B	33%	ABC
Australis	2	78	B	29.5	B	30%	BC
USDA 1-2	2	8	C	27.1	B	23%	C

^1^ Indicates the first or second experimental trial. ^2^ Area under the insect growth curve (AIGC) similar to calculated aphid days, based on the area under the disease progress curve formula from Shaner and Finney [47,48]. ^3^ Average frequency for a cultivar of predatory insects or arachnids or evidence of them present. ^4^ Numbers followed by the same letter within a column are not significantly different (*p =* 0.05).

**Table 4 insects-11-00086-t004:** Population growth of vine mealybug and presence of ants on grapevines evaluated in cultivar study.

Cultivar	Immature Mealybugs AIGC ^1^		Adult Mealybugs AIGC ^2^		Mealybug Ovisacs AIGC		Ant AIGC		Plant Health
Autumn King	530.4	ab ^3^	547.6	a	98.8	ab	53.7	a	4.4
Cabernet Sauvignon	542.5	abc	279.3	ab	56	b	46.9	a	3.4
IAC 572	75.6	c	54.6	b	9.1	b	17.5	c	4.8
RS-3	7.0	c	9.8	b	1.4	b	2.8	c	3.2
Flame Seedless	95.2	abc	133.0	ab	30.8	b	5.6	bc	4.8
Chardonnay	1463	a	532.7	a	161.7	a	39.2	a	4.0
Valley Pearl	100.8	c	272.3	ab	32.9	b	37.1	ab	3.4

^1^ Area under the insect growth curve based on the formula from Shaner and Finney [48]. ^2^ Adult female mealybugs and third instar juveniles. ^3^ Numbers followed by the same letter within a column are not significantly different (*p =* 0.05).

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
