# Peer review of "Identification of Vitis Cultivars, Rootstocks, and Species Expressing Resistance to a Planococcus Mealybug"

_insects, 2020, doi:10.3390/insects11020086_

Round 1

Reviewer 1 Report

It is always interesting to see how scientists deal with unexpected problems in their field experiments. In this particular study, the experiments designed to document plant resistance were impacted by the presence of ants and biocontrol agents. 

1. The presence of ants apparently was not expected, or at least not considered in the design of the experiments. Why would you set up an experiment that included ants with the ants coming from a single location (line 105) somewhere off to one side of the field setup (the distance was not specified)? Some plants were almost certainly much closer to the ant colony than others. Unfortunately, both experiments were compromised by the presence of the ants. If the purpose of the study was to compare germplasm or cultivar suitability for a particular mealybug species, then the results are confounded by the presence of ants (which were more common on plant lines with more mealybugs). The authors essentially assumed that there was no effect of the ants on the results, yet they found substantially different numbers of ants on root stocks than on cultivar lines, and on plants with more mealybugs. 

This leads to the question ‘Were the plant lines in the germplasm experiment arranged in a randomized complete block design’?  This was not stated in the methodology. If this was not done then the experiment was probably further confounded by the fact there was only one ant colony in the experiment. 

2. What was the mesh size on the cage used for the germplasm experiment (see line 91)? Since mobile immatures were produced within the 12 week period, was there any chance that the mealybugs could move between plants?

3. What is the mesh on the ‘paint strainer’ bags in the cultivar preference test? The presence of large numbers of ants and evidence of predators suggests that the bags did not stop ants or predators from entering the bags. If the bags were sealed properly, then no predators should have been able to access the mealybugs and no mealybugs could have escaped the plants.  Since they were not apparently sealed, the bags may have allowed aphids or other pests to attack the plants.  Were any other insects found during the surveys?  Were any plant pathogens observed (such as those transmitted by aphids or other insects)?  This could have had an effect on the results documenting ant populations as well as overall plant health. In addition, the presence of biocontrol agents might have been influenced by the presence of bags.  Some biocontrol agents may not have been able to enter the bags as readily as some others, thus skewing the results. Also, reporting the ‘presence’ of biocontrol agents (parasitized or partly eaten mealybugs) does not provide useful detail on what proportion of the mealybug population may have been removed. Without more detailed information on the effect of the beneficials on the mealybug populations, the data on total mealybug populations developing on various plant lines cannot be attributed to germplasm effects alone.

4. Line 150: the 1 to 5 scale used to measure plant health is not descriptive enough to allow a researcher to repeat the work. For example, how did a ranking of ‘2’ differ from a ranking of ‘3’?More importantly, how does 4.0 differ from 4.6 as seen in Table 4. Creating ranking systems is not simple.  Various statistical results can be obtained depending on how categories are chosen.  The rationale for choosing the categories of plant health needs considerable clarification.

5. In Table 4: under the heading “Plant Health”: It is not clear why the cultivar ‘Valley Pearl’, with a plant health rating of 4.0, is a ‘b’ (significantly different from the 4.6-4.9 rankings given an ‘a’), but Chardonnay with a ranking of 3.8 is given an ‘ab’ and is not significantly different from those with the 4.6-4.9 rankings.  Perhaps this could be rechecked. I suspect that the analysis of the categorical plant health data should have been conducted using a non-parametric test rather an LSMeans test.

6. How many mealybugs were placed on each plant?The authors stated that “The number of mealybugs placed onto each plant, was estimated by the number of first instars that hatchedper ovisacfrom 20randomly selected egg sacs” (lines 100-102). However, the numbers that hatched (along with some estimate of the variability) was not presented.

7. I suspect the authors standardized the plant sizes and number of leaves etc. This should be presented in the methods section. Also, were the insects placed on the plants known to be free of any plant pathogens?  If the mealybugs carried a plant pathogen, then this needs to be reported as the presence of pathogens can impact plant resistance and plant health.

Overall this is a well written study that suffers from a questionable experimental design.  This single year study provides a good pilot study to help determine how a correct experiment should be conducted.  For a future experiment focused on plant resistance, the ants, biocontrol agents and other potential pests should be eliminated.  Once any potential resistance has been documented, then the potential interactive effects of ant presence, biocontrol agents, or plant pathogens could be investigated.  As it stands now, I am not convinced that the potential effects of each factor (plant resistance, ants, biocontrol agents, and other possible pests) can be definitively stated. This is unfortunate because the initial purpose of the experiment, evaluating resistance in various plant lines to Planococcus ficus, has real merit.

Author Response

The presence of ants apparently was not expected, or at least not considered in the design of the experiments. Why would you set up an experiment that included ants with the ants coming from a single location (line 105) somewhere off to one side of the field setup (the distance was not specified)? At the time the experiment was set up, we were not aware of the presence of the ants. It was not until after the experiment was in place and I spoke with the person who we had borrowed the space from that I was told about the ants. Some plants were almost certainly much closer to the ant colony than others. The text has been updated to indicate that the plants were placed in a complete randomized design. It is true that some plants were placed closer to the ant colony than others, but as cultivars were randomly distributed, if there was a random colonization of ants or mealybugs of any given cultivar we should not have seen the drastic cultivar-specific differences that we did. Unfortunately, both experiments were compromised by the presence of the ants. Ant presence is a common occurrence in vineyards, and if resistant rootstocks can be overcome by the added effect of ants tending the mealybugs then they have no use in a breeding program. If the purpose of the study was to compare germplasm or cultivar suitability for a particular mealybug species, then the results are confounded by the presence of ants (which were more common on plant lines with more mealybugs). While it is possible that the ants may have amplified the differences that we observed, if there were no cultivar differences then we should have seen equal distribution of ants and mealybugs in our study. The authors essentially assumed that there was no effect of the ants on the results, yet they found substantially different numbers of ants on root stocks than on cultivar lines, and on plants with more mealybugs. Not necessarily true. We had varying ants between the two cages and between the two experiments. While the ants may be causing an effect, the trends were the same between the two experiments.

This leads to the question ‘Were the plant lines in the germplasm experiment arranged in a randomized complete block design’?  They were so arranged and this has been added to the methodology. This was not stated in the methodology. If this was not done then the experiment was probably further confounded by the fact there was only one ant colony in the experiment. 

What was the mesh size on the cage used for the germplasm experiment (see line 91)? Since mobile immatures were produced within the 12 week period, was there any chance that the mealybugs could move between plants? The entire experiment was within a cage, and there was no screen separating plants in experiment one. Yes, it is perfectly likely that mealybug crawlers were moving between plants. What is the mesh on the ‘paint strainer’ bags in the cultivar preference test? The presence of large numbers of ants and evidence of predators suggests that the bags did not stop ants or predators from entering the bags. If the bags were sealed properly, then no predators should have been able to access the mealybugs and no mealybugs could have escaped the plants. In the second experiment there was not a large number of predators detected. The bags were originally sealed, but the ants chewed through the bags or crawled up through the bottom of the pots. Since they were not apparently sealed, the bags may have allowed aphids or other pests to attack the plants. The plants were checked for other insects, and if anything was detected it was removed. This is a field-based study, and so other biotic factors are to be expected. Were any other insects found during the surveys?  Were any plant pathogens observed (such as those transmitted by aphids or other insects)?  The only plant pathogens detected were those such as sooty mold that grow on honeydew. This could have had an effect on the results documenting ant populations as well as overall plant health. In addition, the presence of biocontrol agents might have been influenced by the presence of bags.  Some biocontrol agents may not have been able to enter the bags as readily as some others, thus skewing the results. It is unclear how biocontrol agents would skew the results if all plants were equally exposed and the only purpose of reporting predators/parasitoids was to note their frequency? Also, reporting the ‘presence’ of biocontrol agents (parasitized or partly eaten mealybugs) does not provide useful detail on what proportion of the mealybug population may have been removed. Without more detailed information on the effect of the beneficials on the mealybug populations, the data on total mealybug populations developing on various plant lines cannot be attributed to germplasm effects alone. The effect of predators on mealybugs is well-documented in other studies. In our work, the point of including predators/parasitoids was to note their frequency and not speculate on the number of mealybugs that had been removed. In the first experiment there was not a bag separation between plants, therefore the pool of biocontrols should be equally available to all plants in the experiment. In the second experiment, any detected predators were removed. Line 150: the 1 to 5 scale used to measure plant health is not descriptive enough to allow a researcher to repeat the work. For example, how did a ranking of ‘2’ differ from a ranking of ‘3’? More importantly, how does 4.0 differ from 4.6 as seen in Table 4. Creating ranking systems is not simple.  Various statistical results can be obtained depending on how categories are chosen.  The rationale for choosing the categories of plant health needs considerable clarification. Additional text has been added to clarify the ranking system, and to state the purpose of making a ranking system. In Table 4: under the heading “Plant Health”: It is not clear why the cultivar ‘Valley Pearl’, with a plant health rating of 4.0, is a ‘b’ (significantly different from the 4.6-4.9 rankings given an ‘a’), but Chardonnay with a ranking of 3.8 is given an ‘ab’ and is not significantly different from those with the 4.6-4.9 rankings.  Perhaps this could be rechecked. I suspect that the analysis of the categorical plant health data should have been conducted using a non-parametric test rather an LSMeans test. We have reran the plant health data using a non-parametric method. The text has been updated to reflect this. How many mealybugs were placed on each plant? The authors stated that “The number of mealybugs placed onto each plant, was estimated by the number of first instars that hatched per ovisac from 20 randomly selected egg sacs” (lines 100-102). However, the numbers that hatched (along with some estimate of the variability) was not presented. We didn’t place mealybugs on each plant, we used ovisacs as our standard metric for the first experiment. The text has been modified to reduce confusion. As previously noted however, mealybug juveniles can move among plants, and in our first experiment the plants were not placed in a fashion to discourage movement. The second experiment used mealybug juvenile numbers as our starting metric. I suspect the authors standardized the plant sizes and number of leaves etc. This should be presented in the methods section. Also, were the insects placed on the plants known to be free of any plant pathogens?  All cuttings were taken from symptomless vines, this has been added to the materials and methods. There are a number of endosymbionts in mealybugs, but the lab colony was not tested for presence of any particular pathogen. The lab colony had been raised on butternut squash for multiple generations. Plants of a similar age and size were used, but each of these cultivars has different levels of vigor. The text has been updated to indicate that plants used were of a similar size and age. If the mealybugs carried a plant pathogen, then this needs to be reported as the presence of pathogens can impact plant resistance and plant health. It is not known if the lab-raised colony had any plant pathogens. 

Overall this is a well written study that suffers from a questionable experimental design.  This single year study provides a good pilot study to help determine how a correct experiment should be conducted.  For a future experiment focused on plant resistance, the ants, biocontrol agents and other potential pests should be eliminated. This work was done in the field, and removing predators/pests and ants from field studies is not reasonable without using insecticides (which could damage the mealybugs) or baits (which were initially used, but had little effect on the ant presence). Once any potential resistance has been documented, then the potential interactive effects of ant presence, biocontrol agents, or plant pathogens could be investigated.  As it stands now, I am not convinced that the potential effects of each factor (plant resistance, ants, biocontrol agents, and other possible pests) can be definitively stated. We are not definitively stating the effects of any given factor, but instead are showing through two separate experiments that certain rootstocks are less favorable for mealybug growth and reproduction. This is unfortunate because the initial purpose of the experiment, evaluating resistance in various plant lines to Planococcus ficus, has real merit.

Reviewer 2 Report

Dear Authors,

This certainly is an interesting paper with promising results. I do however have some issues with the presentation which is not always clear enough to understand. I think the papers presentation can be considerably improved based on available experimental data and conclusions could be much clearer after re-analysis of available data.

Here are a number of suggestions:

General:

Language:

Punctuation not always very obvious. (example: Page 1, line 34. Skip comma after 'in part', add comma after [7]. Phrases not always clear. For example: Page 1 lines 30-31 "with the exception of Phylloxera' woul be better placed AFTER end of phrase (arthropods pests). Spelling: Page 1 line 41 growerS terminology: Some terminology is not clear so should be avoided, examples: 'sustainable' (page 2 line 65). No definition given or available. IPM systems are integral for mealybug (P2 L55). I do not understand this

intro:

Figure 1: In the legend of the figure, lines 53 and 54 there is text about almonds ??? that should be deleted.

Life cycle might ned some more precision, allowing you to explain why rootstocks were tested and might be interesting (Mealybugs living on roots in winter)   

M&M:

L85: Was this material virus free? It could be that plants are virius infested and that might change their suitability for mealybugs?

L91 The setup on trial 1 (repeated twice) is NOT clear. What is a screened cage? Size, mesh size, Beneficials can come in or not? What climatic conditions were there? What time of the year ?

L102: Size of egg clusters in ovisac: Please provide data for 20 random sacs to be able to see if here is much variation. With only two egg sacs per plant....will there be variation?

L106: You mention a suite of possible beneficials. Which ones were actually recorded?

L111: If I understand this correctly you are calculating mealybug-days here. This is current practice for entomology (aphid-days).

L116 The formula is wrong for some sub- and superscripts (Yi+1); Ti+1)    the x is a multiplier (?), better use * ?

L120: 'compared' ??? I think it is simply multiplied and then summed ???

L84 and 132: Why do you talk about 'germplasm evaluation' and 'Cultivar preference' whereas trials are in fact very similar? Preference would imply a CHOICE test, this is not the case.

In trial 2.2 the fabric excluded beneficials? However the ants still came in ? Please explain. How did this influence the results?  

Results:

Line 166: You talk about mealybug and ant population growth.
For mealybugs there is growth only after initial infestation. Your data do show total population 'pressure' but not growth.  For ANTS this strictly speaking is not a growing population since no geographical limit is defined....and they come from outside the trial Low 'accumulated mealybug days' can be due to low success of installation/inoculation (antixenosis?) or low increase in populations (antibiosis?).

Your data should be able to distinguish between these two cases.

Please present separately to improve interpretation. The difference between the two replicates of trial 1 are very big. Can you explain why? Could it be that a pb occurred during infestation or rather during pop. build up? 

Table 3: Remove horizontal line below 10-17a Same for table 4 below AUTUMN KING ?

Discussion

Though you state these are semi-field / cage trials, you do not discuss the impact this can have on the results.

Please explain why you think 'resistant' rootstocks can have an effect of populations on cultivars grafted on them...

Line 225 anti-biosis ==> antibiosis

Author Response

Punctuation not always very obvious. (example: Page 1, line 34. Skip comma after 'in part', add comma after [7]. Phrases not always clear. Fixed in the text. For example: Page 1 lines 30-31 "with the exception of Phylloxera' woul be better placed AFTER end of phrase (arthropods pests). Fixed in the text.  Spelling: Page 1 line 41 growerS terminology: Fixed in text. Some terminology is not clear so should be avoided, examples: 'sustainable' (page 2 line 65). No definition given or available. Removed from text to reduce confusion. IPM systems are integral for mealybug (P2 L55). I do not understand this The term is defined in the same sentence? Are you stating that you do not understand what an integrated pest management system is?

intro:

Figure 1: In the legend of the figure, lines 53 and 54 there is text about almonds ??? that should be deleted. Text was deleted

Life cycle might ned some more precision, allowing you to explain why rootstocks were tested and might be interesting (Mealybugs living on roots in winter)   

M&M:

L85: Was this material virus free? It could be that plants are virius infested and that might change their suitability for mealybugs? Material was collected from non-symptomatic vines. However, vines were not virus tested prior to experiment.

L91 The setup on trial 1 (repeated twice) is NOT clear. What is a screened cage? Size, mesh size, Beneficials can come in or not? Beneficials can come in, which is why their presence was measured. What climatic conditions were there? What time of the year ? This is an outdoor cage in the summer in the San Joaquin Valley, it was hot and dry. The text has been modified to include the month and year the study was conducted.

L102: Size of egg clusters in ovisac: Please provide data for 20 random sacs to be able to see if here is much variation. With only two egg sacs per plant....will there be variation? Text regarding the average number of insects per eggsacs has been removed as it created confusion. We placed two eggsacs onto each plant, and that was our metric for standardizing the number of mealybugs per plant in the first experiment.

L106: You mention a suite of possible beneficials. Which ones were actually recorded? We recorded presence only as lacewing eggs, spiders, various robber flies, and mealybug husks with holes in them were all detected across the plants in experiment 1.

L111: If I understand this correctly you are calculating mealybug-days here. This is current practice for entomology (aphid-days). I am not familiar with aphid days, but based on my quick googling of the topic, it seems that we would need to include the doubling time into the calculator. The calculation used does not include generation time.

L116 The formula is wrong for some sub- and superscripts (Yi+1); Ti+1)    the x is a multiplier (?), better use * ? This is a formatting issue, and it looks correct on our computer, however it may have changed during journal formatting. I have included it as a PDF image.

L120: 'compared' ??? I think it is simply multiplied and then summed ??? The text has been edited to reflect that it should be “multiplied”.

L84 and 132: Why do you talk about 'germplasm evaluation' and 'Cultivar preference' whereas trials are in fact very similar? Preference would imply a CHOICE test, this is not the case. We chose the names to differentiate the two trials, for clarification we have changed the name of the second one to Cultivar evaluation.

In trial 2.2 the fabric excluded beneficials? However the ants still came in ? Please explain. How did this influence the results?  The ants chewed through the netting in some cases and burrowed through the bottom of the pots in others. Beneficials, when detected, were removed.

Results:

Line 166: You talk about mealybug and ant population growth. 
For mealybugs there is growth only after initial infestation. Your data do show total population 'pressure' but not growth.  For ANTS this strictly speaking is not a growing population since no geographical limit is defined....and they come from outside the trial Low 'accumulated mealybug days' can be due to low success of installation/inoculation (antixenosis?) or low increase in populations (antibiosis?). The text has been edited to reflect that we are measuring ant density and not population growth.

Your data should be able to distinguish between these two cases. It is possible that crawlers moved between plants (either directly or moved by ants), therefore we can not say with confidence that we are observing antixenosis or antibiosis. Based on the results in the second trial, it appears that we have both types of resistance mechanisms deployed in different rootstocks. However, we are working on additional experiments to determine this.

Please present separately to improve interpretation. The difference between the two replicates of trial 1 are very big. Data from the two replicates of trial one are clearly distinguished in Table 3, were you requesting a separate table for each? If so, the authors would politely disagree that this would improve interpretation, because a large part of what is being shown is that the resistance trends are the same whether you have high or low mealybug infestations. Can you explain why? Could it be that a pb occurred during infestation or rather during pop. build up?  The differences could be due to a number of things, but may be because we had reduced establishment, and there were fewer ants in the second replicate of trial 1.

Table 3: Remove horizontal line below 10-17a Same for table 4 below AUTUMN KING ? Removed.

Discussion

Though you state these are semi-field / cage trials, you do not discuss the impact this can have on the results. We have added several statements about our experimental setup, acknowledging some of the limitations, but also emphasizing that these conditions are more similar to actual vineyard conditions.

Please explain why you think 'resistant' rootstocks can have an effect of populations on cultivars grafted on them... Text was added to explain why resistant rootstocks could negatively affect mealybug populations.

Line 225 anti-biosis ==> antibiosis The text has been fixed.

Round 2

Reviewer 1 Report

I sympathize with the authors situation. I also recognize that field work can be difficult when unexpected factors cause confounding problems.  In this case the authors claim that the experimental design was resilient to any problems from uncontrolled and variable ant populations and uncontrolled and variable beneficial insect populations.  I am not convinced. 

The authors claim ants were not a problem because ants occur in many vineyards and any cultivar or rootstock that is not resistant to mealybugs in the presence of ants would be useless. First, documenting mealybug resistant cultivars or rootstock in the absence of ants has substantial value. These lines could then be used for breeding purposes or a grower could use them with the understanding that ant control would be required.  This information is lost in this study because ants “may have amplified the differences we observed”. 

This is a single year study.Nearly all germplasm comparisons require at least two years of field data to show that the results can be repeated.  The similar “trend” found in a second replicate that was conducted using a different cage design was essentially a different experimental design.

It is not clear why the authors think that variable numbers of natural enemies (that reached different levels of percent infestation on specific cultivars) would not potentially affect numbers of mealybugs. Why cage the plants at all if beneficials were not considered a problem?  One could argue that, like the ants in vineyards, beneficials occur in most vineyards.  The problem is that the uncounted beneficial insects could impact mealybug populations in such a way as to slow their growth and development.  Mealybug growth and development was the key factor that was measured to determine potential resistance of the germplasm lines tested.  If 1) the biocontrol agents preferred one of the test lines over another (see many examples in Vet and Dicke. 1992. Ann Rev. Entomology 37:141-72; Heinz and Parrella 1994. Biological Control 4: 305-318; Mody et al. 2017. Agric., Ecosyst., Environ. 245: 74-82), 2) if the presence of larger numbers of mealybugs on one line was more attractive to the beneficials and they preferentially attacked the lines with higher numbers of mealybugs and subsequently removed more of these, or 3) if the variable caging techniques were more favorable to one beneficial species over others, then the results are almost certainly compromised.  Unfortunately, with the current experimental design, the influence of these confounding factors cannot be documented. If one or more of the lines was more attractive to beneficials (see references above), this information would be of considerable value.

One of the basic tenants of science is that the work has to be repeatable.The authors indicated that they initially sampled the offspring from 20 egg sacs to get an estimate of how many crawlers were placed on each plant when two egg sacs/plant were used.  To be repeatable we need to know how many eggs were in an ‘average’ egg sac and how many likely hatched (preferable with some measure of variability).

I stopped at this point.  I appreciate that the authors corrected some of the statistical analyses and included information that the test was conducted using a randomized complete block design. However, in my opinon, the experimental design is too flawed and incomplete to justify publication.

Author Response

The authors claim ants were not a problem because ants occur in many vineyards and any cultivar or rootstock that is not resistant to mealybugs in the presence of ants would be useless. First, documenting mealybug resistant cultivars or rootstock in the absence of ants has substantial value. These lines could then be used for breeding purposes or a grower could use them with the understanding that ant control would be required.  This information is lost in this study because ants “may have amplified the differences we observed”.  We would respectfully disagree that the information is “lost”. As with any experiment, there can be improvements to design and these can be taken into account for future work, and we agree with the reviewer that it would have been preferable not to have ants or predators. Certainly at the start of the experiment we did not intend to have substantial numbers of ants and predators. However, these are aspects that can affect host resistance efficacy in vineyard settings, and as they were detected in the study, we felt it was necessary to include this information within this paper to provide an accurate reflection of the potential usefulness of these materials. This paper is intended as a starting point for continued research into mealybug host resistance.

This is a single year study. Nearly all germplasm comparisons require at least two years of field data to show that the results can be repeated.  The similar “trend” found in a second replicate that was conducted using a different cage design was essentially a different experimental design. Perhaps we were unclear. The results presented are based on one replicated study, the germplasm evaluation in two replicated separate cages in 2017 and an unreplicated study in 2018 showing a similar trend as the 2017 data using a different experimental design with overlapping genotypes. This means that we have evidence of resistance in one replicated experiment, and evidence of resistance/favorability for mealybugs in an entirely separate (e.g. different experimental design) study. Whether we had high mealybug pressure (cage 1 and unreplicated cultivar evaluation) or moderate mealybug pressure (cage 2), the results were similar. This is what we meant when we said a similar “trend” was found.

It is not clear why the authors think that variable numbers of natural enemies (that reached different levels of percent infestation on specific cultivars) would not potentially affect numbers of mealybugs. The beneficials had equal access to each plant, and each plant had the same number of mealybugs to start. Why cage the plants at all if beneficials were not considered a problem? Plants were caged to keep them in a confined and easily identifiable space in the field, beneficials did not appear in large numbers on any plant. One could argue that, like the ants in vineyards, beneficials occur in most vineyards.  This is true, and beneficials are highly desirable. The problem is that the uncounted beneficial insects could impact mealybug populations in such a way as to slow their growth and development. See previous comment about beneficial effects being randomly distributed. We count the total number of beneficials in this study, as we did not observe more than 1 or 2 on a single plant, and their presence wasn’t consistent. If we had noticed large differences in the number of predators we would have recorded that data, as we did with the ants, because the goal of this work is to provide a first look at potential mealybug host resistance in grape. However, upon evaluation of the text we have added new verbiage to clarify that the predator numbers represented are the average frequency of detecting a beneficial on any given cultivar. We also added in a correlation analyses looking at the association of beneficial detection with mealybug presence. Mealybug growth and development was the key factor that was measured to determine potential resistance of the germplasm lines tested.  If 1) the biocontrol agents preferred one of the test lines over another (see many examples in Vet and Dicke. 1992. Ann Rev. Entomology 37:141-72; Heinz and Parrella 1994. Biological Control 4: 305-318; Mody et al. 2017. Agric., Ecosyst., Environ. 245: 74-82), If biologicals prefer one cultivar to another that would indeed be beneficial to know, however in this study detected beneficials were low (~0-2 per plant, and the frequency with which they were detected across the 12 weeks of the study were also low). One could also argue that attracting beneficials could be useful as a source of resistance to breed into new cultivars, which could be an additional type of resistance to evaluate moving forward. We have included a sentence in the discussion to this point. 2) if the presence of larger numbers of mealybugs on one line was more attractive to the beneficials and they preferentially attacked the lines with higher numbers of mealybugs and subsequently removed more of these, If beneficials were attracted to plants with higher numbers of mealybugs then we would have seen a negative correlation with mealybugs if the beneficials were reducing mealybug , which we did not or 3) if the variable caging techniques were more favorable to one beneficial species over others, then the results are almost certainly compromised. We did not have variable caging technicques within an experiment, so it is unclear to what the reviewer is referring. Unfortunately, with the current experimental design, the influence of these confounding factors cannot be documented. If one or more of the lines was more attractive to beneficials (see references above), this information would be of considerable value. We agree that it would be beneficial. However, that was not a part of the current study, but may be of interest for future study by other researchers.

One of the basic tenants of science is that the work has to be repeatable.The authors indicated that they initially sampled the offspring from 20 egg sacs to get an estimate of how many crawlers were placed on each plant when two egg sacs/plant were used. The reference to estimated number of eggs is no longer in the manuscript, as of the previous set of revisions. To be repeatable we need to know how many eggs were in an ‘average’ egg sac and how many likely hatched (preferable with some measure of variability). Planococcus ficus ovisacs typically contain between 50-500 eggs, depending on host plant and temperature; however, the resulting number of live crawlers (1st instars) also depends on host plant condition and temperature (Daane, unpubl data) and an accurate count would on the plant would require a search and partial destruction of the vine. 

I stopped at this point.  I appreciate that the authors corrected some of the statistical analyses and included information that the test was conducted using a randomized complete block design. However, in my opinon, the experimental design is too flawed and incomplete to justify publication. 

Reviewer 2 Report

After re-reading the revised paper and the response to my own review and that of reviewer one, I have to say that I am disappointed.

The authors have certainly made a number of modifications that were requested but not on some of the more essential points:

-  They have not provided information(that they have) on the difference in the establishment between the two germplasm evaluation trials

-  If they would distinguish between initial establishment (N0) and subsequent growth (r values, the real 'trend' in the population) (data that they have in the dataset !!!!) the study would be much more valuable to compare germplasm (and less critique would be given to the unwanted interaction with aphids and beneficials)

- In spite of both reviewers asking for clarification on numbers of eggs in egg-sacs (that can then be compared to quantify installation success) they do not reply to this.

- Instead they still present a single statistical analysis for all data using two 'replicates' that are clearly NOT identical, so they should be analyzed separately.

- They claim their methodology for calculating population pressure is not similar to 'aphid-day' calculation whereas they are (and generation time is not required as an input). This seems very strange for an entomological publication.

So..... I repeat what I said before (and reviewer 1 seems to agree):

- The subject of the paper and aim of the study a certainly valid.

- The data can and should be better presented and analyzed.

- The paper has not been improved sufficiently, the effort shown by the authors is not enough.

Author Response

After re-reading the revised paper and the response to my own review and that of reviewer one, I have to say that I am disappointed.

The authors have certainly made a number of modifications that were requested but not on some of the more essential points:

-  They have not provided information(that they have) on the difference in the establishment between the two germplasm evaluation trials. 

-  If they would distinguish between initial establishment (N0) and subsequent growth (r values, the real 'trend' in the population) (data that they have in the dataset !!!!) the study would be much more valuable to compare germplasm (and less critique would be given to the unwanted interaction with aphids and beneficials) As indicated to reviewer 1, Planococcus ficus ovisacs contain between 50-500 eggs, depending on host plant and temperature; however, the resulting number of live crawlers (1st instars) also depends on host plant condition and temperature (Daane, unpubl data) and an accurate count would on the plant would require a search and partial destruction of the vine. If the reviewer so requests we could provide a supplemental table with actual count data for juveniles. However, these data would not provide the information that the reviewer is searching for in that juvenile values fluctuated widely even on a single plant. For example, in one vine on which increasing numbers of adults and 3rd stage juveniles were identified, juvenile counts fluctuated between our two-week rating periods from 5, 19, 6, 21, 3, 2, and 2 (on the final count date). This is, in part, because we are evaluating each plant for 1 minute. Juveniles are quite small and tend to hide in crevices. This is not a lack of "effort" on the authors' part, this is a limitation of the system. 

- In spite of both reviewers asking for clarification on numbers of eggs in egg-sacs (that can then be compared to quantify installation success) they do not reply to this. Please see above comment regarding average number of eggs per sac and hatching.

- Instead they still present a single statistical analysis for all data using two 'replicates' that are clearly NOT identical, so they should be analyzed separately. It is unclear how the data were not analyzed separately. The data in Table 3 shows a value and corresponding letter for each line in respect to each cage. We used standard statistical analyses to slice the interaction to determine what the effect for each line within each cage was. However, as per the reviewer’s request, we have also ran the data separately, see attached supplementary file (Table 1). However, we are not sure what value this provides beyond what is shown in Table 3.

- They claim their methodology for calculating population pressure is not similar to 'aphid-day' calculation whereas they are (and generation time is not required as an input). This seems very strange for an entomological publication. This sentence is not clear. What does the reviewer mean by “whereas they are”? We are also unsure what the benefit of including an aphid day calculator would be? The aphid days calculator is to predict the total number of aphids on a plant, not provide actual counts. It also assumes a stagnant doubling time across the time period. Because we are using different cultivars, and one of the possible effects of these cultivars is an effect on development, it would be inaccurate to assume a static doubling time. This is also true in that temperature can affect development time, and these studies were conducted from June to September where we have large differences in temperature.

So..... I repeat what I said before (and reviewer 1 seems to agree):

- The subject of the paper and aim of the study a certainly valid.

- The data can and should be better presented and analyzed.

- The paper has not been improved sufficiently, the effort shown by the authors is not enough. We regret that the reviewer feels that we have not put forth a great enough effort. While we would be happy to include additional data that we do have, we cannot provide data that we do not have. These new and exciting results are meant to show that there is potential resistance to mealybug in grape, and serve as a starting point for further work. We agree with both reviewers that more work is needed to determine how ants, beneficials and mealybugs interact to affect host resistance. However, this was not the aim or focus of this paper.

Round 3

Reviewer 2 Report

For ease I just copy parts of my second review (blue) and your answers to those (green) and my third review (normal text)

-  If they would distinguish between initial establishment (N0) and subsequent growth (r values, the real 'trend' in the population) (data that they have in the dataset !!!!) the study would be much more valuable to compare germplasm (and less critique would be given to the unwanted interaction with aphids and beneficials) As indicated to reviewer 1, Planococcus ficus ovisacs contain between 50-500 eggs, depending on host plant and temperature; however, the resulting number of live crawlers (1st instars) also depends on host plant condition and temperature (Daane, unpubl data) and an accurate count would on the plant would require a search and partial destruction of the vine. If the reviewer so requests we could provide a supplemental table with actual count data for juveniles. However, these data would not provide the information that the reviewer is searching for in that juvenile values fluctuated widely even on a single plant. For example, in one vine on which increasing numbers of adults and 3rd stage juveniles were identified, juvenile counts fluctuated between our two-week rating periods from 5, 19, 6, 21, 3, 2, and 2 (on the final count date). This is, in part, because we are evaluating each plant for 1 minute. Juveniles are quite small and tend to hide in crevices. This is not a lack of "effort" on the authors' part, this is a limitation of the system. 

Ovisac egg numbers and results of individual counts:
OK, thank you for this explication. So the population estimate is based on fixed time-sampling, and authors indicate crawlers and early stages are very difficult to detect. This will cause a bias for observation between the easily detectable later stages and the early stages (that supposedly are very numerous, more than the following stages?). It would certainly have been better to explain this.

So based on this: How accurate is the actual population estimate and –as a consequence- the accuracy of the AUDPC you use?

It might have been better to focus only on the larger (easy to detect) stages and have a more exhaustive count of these?

- Instead they still present a single statistical analysis for all data using two 'replicates' that are clearly NOT identical, so they should be analyzed separately. It is unclear how the data were not analyzed separately. The data in Table 3 shows a value and corresponding letter for each line in respect to each cage. We used standard statistical analyses to slice the interaction to determine what the effect for each line within each cage was. However, as per the reviewer’s request, we have also ran the data separately, see attached supplementary file (Table 1). However, we are not sure what value this provides beyond what is shown in Table 3.

Legend of Table 3 says “4 Numbers followed by the same letter within a column are not significantly different (P = 0.05)”. Columns contain data from trials 1 and 2 in alternate lines. So I interpret this as a single statistical analysis done on both trials.

The additional table seems correct but please do indicate that trails were analysed separately so do not use the same letters (or use a different font) in a single column. Maybe better to divide table in two parts, one for each trial.   

Because of separating the trials in the new table we can now see easily that there are (in each trial) only two different ‘homogenous groups” (A and B). In the previous table (still in the manuscript), the data suggest THREE different groups (cd,a,e,de,cd,e) for trial 2.      

- They claim their methodology for calculating population pressure is not similar to 'aphid-day' calculation whereas they are (and generation time is not required as an input). This seems very strange for an entomological publication. This sentence is not clear. What does the reviewer mean by “whereas they are”? We are also unsure what the benefit of including an aphid day calculator would be? The aphid days calculator is to predict the total number of aphids on a plant, not provide actual counts. It also assumes a stagnant doubling time across the time period. Because we are using different cultivars, and one of the possible effects of these cultivars is an effect on development, it would be inaccurate to assume a static doubling time. This is also true in that temperature can affect development time, and these studies were conducted from June to September where we have large differences in temperature.

Cumulative aphid-days are used extensively: see for example  Hodgson, Erin W.; VanNostrand, Gregory R.; and Rusk, Ryan, "Soybean Aphid Efficacy Evaluation" (2012). Iowa State Research Farm Progress Reports. 145. http://lib.dr.iastate.edu/farms_reports/145. Formula is included and explained, and strictly identical to the AUDPC formula

Author Response

We have taken the file you provided and highlighted our new responses in yellow for ease of identification. 
